# Assessing the reliability of phone surveys to measure reproductive, maternal and child health knowledge among pregnant women in rural India: a feasibility study

Angela Ng [1], Diwakar Mohan [1], Neha Shah [1], Kerry Scott [1], Osama Ummer [2], Sara Chamberlain [3,4], Aarushi Bhatnagar,[5] Diva Dhar,[6] Smisha Agarwal,[1] Rajani Ved,[7,8] Amnesty Elizabeth LeFevre [9], Kilkari Impact Evaluation Team

For numbered affiliations see end of article.

**Correspondence to**
Angela Ng;
angelang1201@gmail.com

## ABSTRACT

**Objectives** Efforts to understand the factors influencing the uptake of reproductive, maternal, newborn, child health and nutrition (RMNCH&N) services in high disease burden low-resource settings have often focused on face-to-face surveys or direct observations of service delivery. Increasing access to mobile phones has led to growing interest in phone surveys as a rapid, low-cost alternatives to face-to-face surveys. We assess determinants of RMNCH&N knowledge among pregnant women with access to phones and examine the reliability of alternative modalities of survey delivery.

**Participants** Women 5–7 months pregnant with access to a phone.

**Setting** Four districts of Madhya Pradesh, India.

**Design** Cross-sectional surveys administered face-to-face and within 2 weeks, the same surveys were repeated among two random subsamples of the original sample: face-to-face (n=205) and caller-attended telephone interviews (n=375). Bivariate analyses, multivariable linear regression, and prevalence and bias-adjusted kappa scores are presented.

**Results** Knowledge scores were low across domains: 52% for maternal nutrition and pregnancy danger signs, 58% for family planning, 47% for essential newborn care, 56% infant and young child feeding, and 58% for infant and young child care. Higher knowledge (≥1 composite score) was associated with older age; higher levels of education and literacy; living in a nuclear family; primary health decision-making; greater attendance in antenatal care and satisfaction with accredited social health activist services. Survey questions had low inter-rater and intermodal reliability (kappa<0.70) with a few exceptions. Questions with the lowest reliability included true/false questions and those with unprompted, multiple response options. Reliability may have been hampered by the sensitivity of the content, lack of privacy, enumerators' and respondents' profile differences, rapport, social desirability bias, and/or enumerator's ability to adequately convey concepts or probe.

## Strengths and limitations of this study

► COVID-19 has accelerated the use of remote data capture strategies, however, few studies have undertaken a rigorous process of tool development inclusive of reliability testing.

► Study findings suggest that phone surveys can be a reliable modality for measuring knowledge among pregnant women in rural India at a population level, but have low reliability at an individual level.

► To optimise reliability, limit the use of questions which are culturally sensitive, or have open-ended or multiresponse options.

► The sample included pregnant women with reported access to a phone during the day; a population arguably more advantaged than women without access to a phone which could limit the generalisability of findings.

**Conclusions** Phone surveys are a reliable modality for generating population-level estimates data about pregnant women's knowledge, however, should not be used for individual-level tracking.

**Trial registration number** NCT03576157.

## INTRODUCTION

COVID-19 has accelerated the demand for remote data capture solutions, including phone surveys. There is a growing importance and reliance on remote data collection including phone surveys (in India and elsewhere) as COVID-19 has impacted the ease and budgets for data collection.[1] Recommendations on questions to consider in determining the suitability of phone surveys to measure varied outcomes are emerging.[2] However, despite an increasing body of literature on methods,[3 4] including sampling,[5 6]

ethics and obtaining consent,[5 7 8] improving response rates and other performance metrics,[9] significant gaps persist in the standards for developing phone survey tools.

Efforts to measure health practice, prevention and careseeking in India have most commonly been done via face-to-face survey methods, including the population-based National Family Health Surveys (NFHS) carried out approximately every 5–7 years.[10 11] In addition to being costly and time-intensive, these population-based surveys are limited by lengthy recall bias.[12] Phone surveys have the potential to be administered more proximally to event of focus, at lower cost, with more minimal resources, and with potentially less burden to the beneficiary.

In 2019–2020, NFHS findings stated that 54% of women have a mobile phone that they themselves use.[13] Growing access to mobile phones at a household level, and more specifically among women, provides an opportunity to develop alternative modalities for the routine collection of data on health behaviours and knowledge. Despite their immense potential, few applications of phone surveys have been reported for measuring health knowledge, behaviours and outcomes in India[11] and even less information is available on the methods influencing the development of these tools. As phone ownership and access increase in India,[14] a reliable phone survey tool would enable timely, routine and low-cost measurement of the population health status, including knowledge, careseeking and practices.[15]

In this manuscript, we outline the process for developing and refining a phone survey tool for the measurement of reproductive, maternal, newborn, child health and nutrition (RMNCH&N) knowledge among pregnant women in four districts of Madhya Pradesh. We start by developing overall and domain-specific RMNCH&N knowledge scores and then identify the determinants of higher knowledge scores. We then assess differences in the reliability of the knowledge questions over different modalities including face-to-face surveys at two time points (hereafter called test–retest) and caller-attended telephone interviews (CATIs; hereafter called phone surveys). This study illuminates gaps in RMNCH&N knowledge among pregnant women, determinants of RMNCH&N knowledge levels and the reliability of phone survey tools compared with face-to-face tools.

## METHODS
### Study setting
Data collection activities were carried out from August to October 2018 in four districts (Hoshangabad, Mandsaur, Rewa and Rajgarh) of Madhya Pradesh. Madhya Pradesh is a state in central India with a population of over 75 million, predominantly Hindu, and based in rural areas.[16] Significant gaps between men and women persist with regard to literacy (82% of men are literate as compared with 59% of women), and only 29% of women report having access to mobile phones that they themselves can use.[17] For nearly all health indicators, Madhya Pradesh

falls below national averages. Despite near universal attendance of at least one antenatal care (ANC) visit during pregnancy, only 53% of women receive services within the first trimester, and 36% receive the recommended four visits.[17] While institutional delivery rates have increased over time, 20% of women deliver outside the formal health sector, and only 18% receive a health check after birth from a trained provider within 2 days of birth.[17] Among children, 51 per 1000 live births will die prior to their first birthday and 65 per 1000 live births will die prior to their fifth birthday.[17] High mortality rates are affected by undernutrition (43% of children are underweight, 42% are stunted), low rates of exclusive breast feeding (58%) and gaps in access to basic health services including immunisation. Where the latter is concerned, only 73% of children 12–23 months have received the recommended three doses of the diphtheria, pertussis (whooping cough), and tetanus (DPT) vaccine.[17]

### Study design and sampling
The target population was women 5–7 months pregnant at randomisation, ≥18 years of age, speak and understand Hindi, and own or have access to a mobile phone during the morning and afternoon. Survey data and sampling technique were captured as part of baseline survey activities conducted as part of the impact evaluation of the Kilkari mobile health intervention described in detail elsewhere.[18] A face-to-face cross-sectional survey was conducted among (n=5095) women with reported access to a mobile phone during the day (Survey Activity A in figure 1). The survey sample size was powered to detect a 5% change in the practice of exclusive breast feeding from baseline to endline (12 months later), assuming an alpha of 0.05, power of 0.80, 20% loss to follow-up and 35% loss of women due to changes in mobile phone numbers (SIM change). Women excluded included BSNL mobile subscribers due to poor network coverage and those who did not provide consent to participate in the trial. Frontline health workers in the study villages and census of villages produced the list of eligible women. The sample was randomised and stratified by gestational age, parity, age of woman and ownership of phone in Stata using the sample command.[19]

Within 2 weeks of the original survey, two separate surveys were conducted on random samples of these women: test–retest (n=205) and phone surveys (n=375). To assess reproducibility in the surveys, stability of the instrument and inter-rater reliability, a subsample of women who completed the main survey was reinterviewed face-to-face to assess the inter-rater reliability of the responses from the main survey (Survey Activity B in figure 1). In order to develop a validated phone survey tool, pregnant women who previously completed a face-to-face interview and were not interviewed in the test–retest phase were interviewed a second time over the phone to assess the intermodal reliability of the responses from the main survey (Survey Activity C in figure 1).

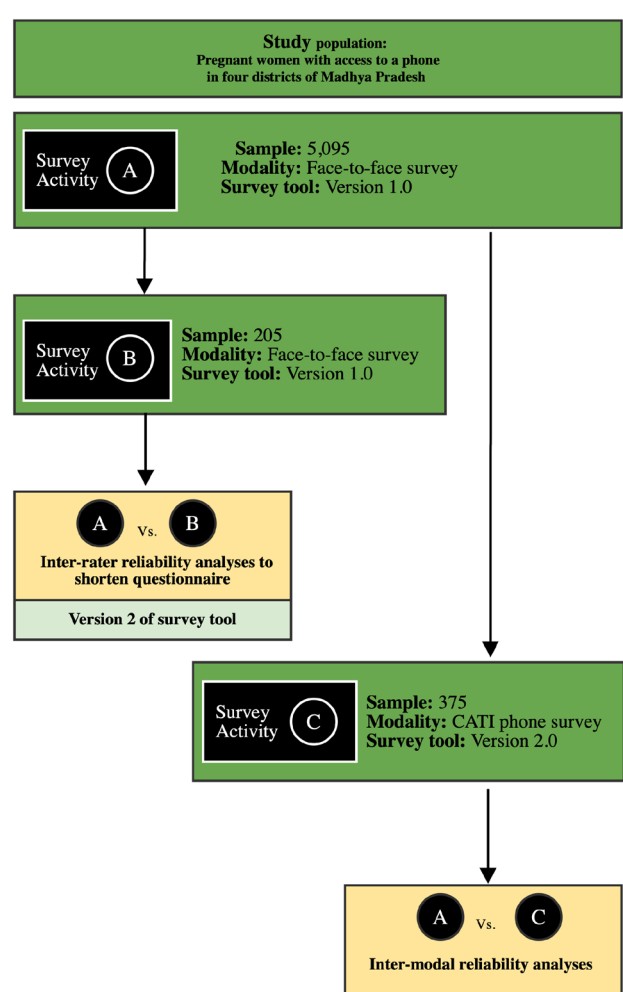

**Figure 1** Completed interviews for face-to-face and CATI surveys. CATI, caller-attended telephone interview.

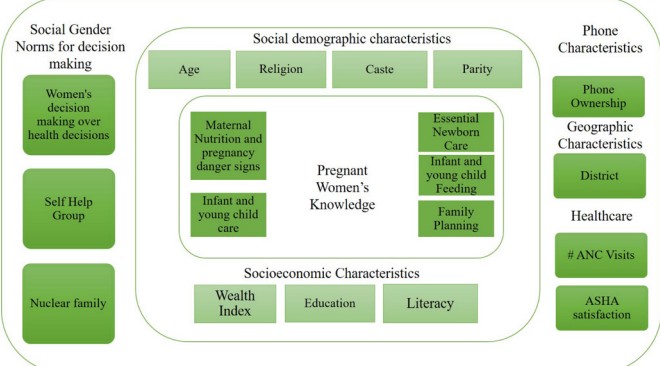

**Figure 2** Conceptual framework for identifying the determinants of pregnant women's RMNCH&N knowledge. ANC, antenatal care; ASHA, accredited social health activist; RMNCH&N, reproductive, maternal, newborn, child health and nutrition.

on knowledge; findings from other modules are reported elsewhere.[18 21]

Female enumerators conducted face-to-face and phone surveys. The phone surveys were conducted out of the National Health Systems Resource Centre in New Delhi. Surveyors had a minimum of high school education and received training on tool content and survey implementation, including informed consent. Supervisors provided day-to-day support and oversight.

## Data analysis

Data were analysed using Stata V.15 (Stata Corporation, College Station, Texas, USA). First, a conceptual framework was developed comprised of factors believed to influence pregnant women's knowledge from previous literature (figure 2).[18–35] At the individual level, characteristics such as phone ownership, healthcare characteristics, socioeconomic and sociodemographic factors were considered. A wealth index was created by dividing the women into five quintiles using principal component analysis. At the community level, social gender norms were analysed such as participation in self-help groups, women being the primary health decision-maker and family living structure as well as geographical characteristics of the community.

In order to best categorise knowledge among pregnant women, survey questions were consolidated into five thematic domains: (1) maternal nutrition and pregnancy danger signs, (2) family planning, (3) essential newborn care, (4) infant and young child feeding, and (5) infant and young child care (online supplemental table 1). For each question, if there was only one correct answer, a binary score was given: 1 if the women knew the correct answer, 0 if not. If there were multiple correct answer options, each correct option was weighted so if all correct options were selected, the final score for that question would be 1. Each question then had equal weight in its category. All the knowledge domains were averaged together at the end to create a final composite score.

Each sample size was calculated assuming a kappa of 0.80, margin of error 0.05, and alpha of 0.5, proportion of positive responses of 0.35 for rater 1 and 0.40 for rater 2: 146 participants were needed. To accommodate a 15% loss to follow-up or refusal for the test–retest, a minimum of 168 women were targeted and ultimately 205 interviewed. Assuming a 20% response rate for the phone survey, a random sample of 880 women were contacted within 1–2 weeks of the initial face-to-face interview, and ultimately 375 completed the interviews.

## Data collection

The survey included seven modules on: (1) mobile phone ownership, use and digital literacy; (2) pregnancy, delivery, child health and family planning knowledge; (3) birth history; (4) experiences during pregnancy care; (5) content of pregnancy care; (6) delivery and postpartum care intentions; and (7) client satisfaction with accredited social health activist (ASHA) services. Survey questions were drawn from standardised national survey tools, the literature and expert judgement and enhanced through cognitive testing and piloting.[18 20] This manuscript focuses

A simple linear regression was performed on the final composite score with the factors thought to have an impact on them. A multiple linear regression was performed to see all together which factors would have an impact on overall women's knowledge. In order to find the best model, only variables with a p value of 0.20 or less during bivariate analyses were used. A multiple linear regression was also performed on each individual knowledge domain to see if particular factors influenced one category more than another.

If the difference in prevalence between survey questions was <15%, kappa scores were used to assess the reliability of the question. Questions were deemed unreliable if the difference in prevalence between survey questions was >15% or if the Cronbach's alpha was 0.7 or higher.[22] Kappa coefficients were adjusted for the difference in prevalence levels using the Prevalence Adjusted Bias Adjusted Kappa score. This was to determine if the questions themselves were reliable, and if so, if the question was reliable over the phone. If the threshold of 0.7 was met for just the test–retest, the question was deemed to be reliable for face-to-face surveys in this context. If the threshold of 0.7 was met for both test–retest and phone survey, the question was deemed reliable for phone surveys in this context.

### Patient and public involvement

Pregnant women were first engaged upon identification in their households as part of a household listing carried out in mid/late 2018. A small number of pregnant women were involved in the refinement of survey tools through qualitative interviews, including cognitive interviews, which were carried out to optimise survey questions, including the language and translation used. COVID-19 and associated travel restrictions prevented further engagement with women interviewed to disseminate study findings.

### RESULTS

#### Pregnant women characteristics

Among 5095 interviewed through the face-to-face survey, 95% were Hindu, 58% were between 18 and 24 years of age, 68% had at least one child, 47% belonged to Other Backward Castes, 8% belong to a self-help group, 41% had completed high school or higher, and 57% were able to read the whole sentence (table 1). A majority of the women interviewed resided in Rewa (57%), followed by Mandsaur, Hoshangabad, and Rajgarh falling at 16%, 8%, and 19%, respectively.

#### Survey reliability

For each survey question, inter-rater reliability was assessed by comparing reported results from face-to-face surveys conducted at two time points (test–retest; n=205), while intermodal reliability was assessed by comparing findings from the face-to-face survey with phone surveys

**Table 1** Characteristics of pregnant women drawn from a face-to-face survey in four districts of Madhya Pradesh

| Characteristic | n | Frequency |
|---|---|---|
| Age | | |
| 18–24 | 2975 | 58.4% |
| 25–34 | 2023 | 39.7% |
| 35+ | 97 | 1.90% |
| Religion | | |
| Hindu | 4848 | 95.2% |
| Muslim | 241 | 4.73% |
| Other | 6 | 0.12% |
| Caste | | |
| General | 1133 | 22.2% |
| Other Backward Caste | 2386 | 46.8% |
| Scheduled Caste/Tribe | 1576 | 30.9% |
| Parity (≥1 child) | 3486 | 68.4% |
| Employed | 1921 | 37.7% |
| Education | | |
| No school | 548 | 10.8% |
| Primary or less | 876 | 17.2% |
| Middle school | 1560 | 30.6% |
| High school | 1666 | 32.7% |
| Higher education | 445 | 8.70% |
| Literacy | | |
| Cannot read at all | 1659 | 32.6% |
| Able to read only parts of sentence | 554 | 10.9% |
| Able to read whole sentence | 2880 | 56.6% |
| Nuclear family structure | 1092 | 21.6% |
| Self-help group | 380 | 7.46% |
| Woman is the primary decision-maker for health decisions | 1451 | 28.5% |
| Attendance of maternity care services | | |
| Antenatal care 1 | 742 | 14.6% |
| Antenatal care 2 | 1225 | 24.0% |
| Antenatal care 3 | 952 | 18.7% |
| Antenatal care 4+ | 2176 | 42.7% |
| Satisfied with the services provided by the ASHA | 4699 | 92.2% |
| Owns phone | 3860 | 75.8% |
| District | | |
| Hoshangabad | 406 | 7.97% |
| Mandsaur | 821 | 16.1% |

Continued

| Table 1 | Continued | |
|---|---|---|
| **Characteristic** | **n** | **Frequency** |
| Rewa | 2920 | 57.3% |
| Rajgarh | 948 | 18.6% |

ASHA, accredited social health activist.

(n=375). Table 2 presents prevalence and reliability estimates by question.

Overall results suggest that population-level estimates of knowledge over time and across modalities were similar, but reliability among individual women in the sample was poor. Prevalence estimates illustrate the latter, demonstrating that women's knowledge across and within domains was low. Questions with low reliability included true/false questions and those with unprompted, multiple response options.

Figure 3 graphs the correlation between the prevalence and bias-adjusted kappa statistics for face-to-face surveys against the in-person retest and the phone survey. The correlation forms a positive linear line, representing a strong positive reliability between the face-to-face and phone survey options.

### Determinants of RMNCH&N knowledge

Figure 4 shows average knowledge scores for the five composite knowledge domains: maternal nutrition and pregnancy danger signs (52%), family planning score (58%), immediate post-birth care for the neonate (47%), infant and young child feeding (56%), and infant and young child care (58%). The average of all domains was used to create a final composite score (54%).

Table 3 presents results from bivariate and multivariate regressions on factors influencing pregnant women's knowledge (n=5095). In the multivariable model, women's knowledge was significantly higher among older women (25–34 years, p<0.001; >34 years, p=0.02); with higher education (p=0.009); able to read a full sentence (p<0.001); living in a nuclear family (p<0.001); who reported being the primary decision-maker in health decisions (p<0.001); had attended a greater number of ANC visits (p<0.001) and reported being satisfied with ASHA services (p<0.001). Women from the rural districts of Mandsaur and Rajgarh had significantly lower knowledge scores than women in the reference periurban district of Hoshangabad. Women's phone ownership, socioeconomic status and caste were not associated with significant differences in knowledge.

Online supplemental table 2 explores determinants of knowledge for each of the five domains. Across all domains, knowledge was significantly higher among older women (25–34 years of age), with a least one child, able to read a whole sentence and who reported being satisfied with services provided by the ASHA. Knowledge of maternal nutrition and pregnancy danger signs was significantly higher among women living in a nuclear family, with higher education and who had attended a greater number of ANC visits. Family planning knowledge was significantly higher among women in the highest socioeconomic strata, with primary school or higher education. Knowledge of immediate post-birth care for the newborns was similarly higher among women in the highest socioeconomic strata and with higher education.

### DISCUSSION

The study sought to develop and refine a phone survey tool for the measurement of RMNCH&N knowledge among pregnant women in four districts of Madhya Pradesh. Findings suggest that there is a correlation between face-to-face and retest reliability and face-to-face and phone reliability. Instances were not observed where there was retest reliability for face-to-face and poor reliability for face-to-face versus phone. Questions with unprompted and more possible response options (eg, foods to eat during pregnancy) had similarly poor inter-rater and intermodal reliability as questions that sought to pin respondents down to one answer (eg, How many tetanus injections should you have during pregnancy?). This suggests that the questions themselves may not be well suited to repeat questioning. Overall knowledge levels were low for most questions asked. This underscores the need to bolster awareness among women and improve access to health information. It too may indicate that some questions and response options used were not culturally appropriate or relevant for this context (eg, alcohol, coffee and cigarettes are uncommonly consumed in rural India by women which may explain the low awareness of these items when asked 'What should you not eat or drink during your pregnancy?'). The collective body of findings suggests that survey estimates may be appropriate for ascertaining population-level estimates rapidly, however, contraindicated in instances where longitudinal tracking or individual-level analyses are required. This may have critical implications for those looking to use remote surveys for measuring intermediate programme outcomes, including knowledge.

Study findings contribute to the limited body of literature on how to optimise the development of phone surveys for use in the measurement of population-level knowledge and health outcomes. Few studies to date have sought to refine phone survey tools through a rigourous process of reliability testing.[23] Findings from a 2017 systematic review identified only 10 studies which have reported using reliability testing to develop phone survey tools; five of these included face-to-face and CATI survey comparisons.[23] In Honduras and Peru,[24] comparisons were made across independent samples, while in Lebanon[25] and two separate surveys in Brazil,[26] comparisons were made using the same sample across two modalities (as in our study).[18 22] Overall findings broadly suggest that there was concordance in the results across CATI and face-to-face modalities.[21] However, for some outcomes, prevalence estimates did differ across modalities.[27] Studies in

**Table 2** Prevalence and reliability estimates for knowledge questions among pregnant women

| Question text | Response options | Baseline prevalence | Retest prevalence | Phone prevalence | Retest kappa | Phone kappa |
|---|---|---|---|---|---|---|
| **Domain: maternal nutrition and pregnancy danger signs** | | | | | | |
| What foods should you eat during pregnancy? | Fish | 0.07 | 0.08 | 0.02 | 0.79 | 0.84 |
| | Meat | 0.08 | 0.06 | 0.04 | 0.80 | 0.83 |
| | Eggs | 0.13 | 0.08 | 0.05 | 0.79 | 0.68 |
| | Milk/dairy products | 0.73 | 0.79 | 0.59 | 0.32 | 0.17 |
| | Fruits | 0.89 | 0.92 | 0.73 | – | – |
| | Green leafy vegetables | 0.94 | 0.93 | 0.76 | – | – |
| | Pulses and nuts | 0.86 | 0.88 | 0.34 | – | – |
| What should you not eat or drink during your pregnancy? | Alcohol | 0.03 | 0.02 | 0.01 | 0.94 | 0.93 |
| | Coffee | 0.03 | 0.01 | 0.05 | 0.95 | 0.84 |
| | Cigarettes | 0.05 | 0.02 | 0.01 | 0.95 | 0.83 |
| | Tea | 0.14 | 0.16 | 0.09 | 0.58 | 0.45 |
| Anaemia (khoon ki kami) in pregnancy can affect the growth and development. What should you do if you are anaemic? | Take IFA tablet daily | 0.64 | 0.73 | 0.56 | 0.30 | 0.21 |
| | Tea | 0.40 | 0.63 | 0.07 | – | – |
| | Cigarettes/bids | 0.83 | 0.76 | 0.34 | – | – |
| How many tetanus injections should you have during pregnancy? | 2 shots | 0.66 | 0.52 | 0.65 | 0.77 | 0.58 |
| What are IFA tablets? | Help prevent/treat anaemia | 0.80 | 0.90 | 0.64 | – | – |
| | Improve the health/well-being of my baby | 0.73 | 0.85 | 0.43 | – | – |
| What danger signs during pregnancy and before labour starts would lead you to go to the health facility immediately? | Yellowing of skin | 0.10 | 0.02 | 0.01 | 0.88 | 0.78 |
| | Vaginal bleeding | 0.31 | 0.32 | 0.10 | – | – |
| | Vaginal discharge | 0.35 | 0.33 | 0.10 | – | – |
| | Convulsions | 0.42 | 0.52 | 0.09 | – | – |
| | Stomach cramps | 0.87 | 0.95 | 0.45 | – | – |
| | Swelling on limbs and face | 0.38 | 0.30 | 0.20 | – | – |
| **Domain: infant and young child feeding** | | | | | | |
| How many times per day should newborns be breast fed? | 9–10 times a day, on demand | 0.94 | 0.94 | N/A | 0.75 | N/A |
| What are some of the benefits of breast feeding? | It helps to maintain space between children | 0.01 | 0.01 | 0.01 | 0.98 | 1.00 |
| | Reduces expenditure on medical care as child will fall sick a fewer number of times | 0.11 | 0.05 | 0.05 | 0.82 | 0.62 |
| | The more the child breast feeds, the more milk will be produced | 0.07 | 0.09 | 0.02 | 0.67 | 0.91 |
| | Promotes mother–baby bonding | 0.09 | 0.01 | 0.27 | – | – |
| | Helps build immunity for your child | 0.72 | 0.71 | 0.57 | – | – |
| | Promotes child growth, wellness | 0.85 | 0.96 | 0.54 | – | – |
| How soon after delivery should you give foods or liquids other than mother's milk? | 6 months | 0.83 | 0.74 | 0.75 | 0.64 | 0.64 |
| | Immediately | 0.28 | 0.64 | 0.15 | – | – |

Continued

**Table 2** Continued

| Question text | Response options | Baseline prevalence | Retest prevalence | Phone prevalence | Retest kappa | Phone kappa |
|---|---|---|---|---|---|---|
| What types of foods should a baby be given after 6 months of age? | Cheese | 0.05 | 0.07 | 0.01 | 0.73 | 0.85 |
| | White potatoes, roots | 0.01 | 0.00 | 0.09 | 0.99 | 0.83 |
| | Ripe mangoes, papayas | 0.05 | 0.10 | 0.05 | 0.71 | 0.82 |
| | Beans, peas, nuts | 0.14 | 0.08 | 0.02 | 0.75 | 0.52 |
| | Bread, roti, grains | 0.17 | 0.08 | 0.14 | 0.55 | 0.54 |
| | Fruits or vegetables | 0.14 | 0.16 | 0.16 | 0.52 | 0.63 |
| | Baby food | 0.42 | 0.39 | 0.47 | 0.05 | 0.07 |
| | Plain water | 0.74 | 0.76 | 0.37 | – | – |
| | Juice | 0.10 | 0.07 | 0.35 | – | – |
| | Lentil broth/soup | 0.18 | 0.14 | 0.73 | – | – |
| | Milk | 0.78 | 0.90 | 0.52 | – | – |
| | Pumpkin, squash | 0.78 | 0.88 | 0.38 | – | – |
| | Solid, semisolid, soft food | 0.27 | 0.45 | 0.00 | – | – |
| | Liver | 0.00 | 0.00 | 0.00 | – | – |
| | Chicken | 0.00 | 0.00 | 0.00 | – | – |
| | Meat | 0.00 | 0.00 | 0.00 | – | – |
| | Eggs | 0.00 | 0.00 | 0.00 | – | – |
| | Dried fish | 0.00 | 0.00 | 0.00 | – | – |
| **Domain: infant and young child care** | | | | | | |
| What danger signs do you know about the newborn after delivery? | Redness | 0.01 | 0.01 | 0.01 | 0.98 | 0.99 |
| | Red eyes | 0.02 | 0.02 | 0.01 | 0.94 | 0.96 |
| | Skin lesions | 0.03 | 0.02 | 0.01 | 0.87 | 0.96 |
| | Blueness of hands | 0.03 | 0.01 | 0.01 | 0.96 | 0.91 |
| | Convulsions | 0.04 | 0.02 | 0.06 | 0.85 | 0.84 |
| | Low birth weight | 0.05 | 0.05 | 0.04 | 0.83 | 0.74 |
| | Lethargy | 0.09 | 0.17 | 0.06 | 0.51 | 0.74 |
| | Difficulty feeding | 0.22 | 0.17 | 0.16 | 0.50 | 0.28 |
| | Yellow colour of skin | 0.17 | 0.23 | 0.04 | 0.50 | 0.61 |
| | Pitched cry | 0.26 | 0.21 | 0.30 | 0.44 | 0.20 |
| | Difficulty breathing | 0.36 | 0.35 | 0.05 | – | – |
| | Baby feels hot or cold to touch | 0.30 | 0.49 | 0.15 | – | – |
| | Fever | 0.83 | 0.93 | 0.34 | – | – |
| | Vomiting | 0.52 | 0.49 | 0.24 | – | – |
| How soon after your baby is born should it receive its first vaccination? | At birth | 0.74 | 0.8 | 0.73 | 0.52 | 0.54 |
| | Within 1 month | 0.09 | 0.11 | 0.09 | – | – |
| | 1–2 months | 0.04 | 0.01 | 0.02 | – | – |
| | 2+ months | 0.02 | 0.01 | 0.02 | – | – |
| What are things you can do to prevent your child from getting diarrhoea? | Give baby safe drinking water >6 months | 0.17 | 0.08 | 0.05 | 0.68 | 0.65 |
| | Exclusively breast feed children <6 months | 0.14 | 0.06 | 0.05 | 0.69 | 0.63 |
| | Cover water and food to avoid flies sitting on it | 0.16 | 0.23 | 0.1 | 0.39 | 0.55 |
| | Safe disposal of stools | 0.09 | 0.11 | 0.12 | 0.57 | 0.49 |
| | Make sure the environment is clean | 0.50 | 0.63 | 0.38 | 0.02 | −0.02 |
| | Wash hands | 0.39 | 0.40 | 0.20 | – | – |

Continued

**Table 2** Continued

| Question text | Response options | Baseline prevalence | Retest prevalence | Phone prevalence | Retest kappa | Phone kappa |
|---|---|---|---|---|---|---|
| What should you give your child to treat diarrhoea? | Intravenous | 0.01 | 0.01 | 0.01 | 0.96 | 0.98 |
| | Antibiotic | 0.03 | 0.00 | 0.01 | 0.96 | 0.95 |
| | Injection | 0.07 | 0.18 | 0.01 | 0.50 | 0.80 |
| | Home remedy | 0.10 | 0.02 | 0.01 | 0.93 | 0.78 |
| | Salt and sugar | 0.06 | 0.04 | 0.13 | 0.86 | 0.71 |
| | ORS+zinc | 0.11 | 0.11 | 0.18 | 0.70 | 0.50 |
| | ORS | 0.40 | 0.48 | 0.27 | 0.38 | 0.24 |
| | Antidiarrhoeals | 0.39 | 0.41 | 0.05 | – | – |
| | Other pill or syrup | 0.62 | 0.69 | 0.38 | – | – |
| What are three critical times for a woman to wash her hands? | After defecation | 0.98 | 0.94 | 0.43 | – | – |
| | Before cooking or handling food | 0.91 | 0.94 | 0.37 | – | – |
| | Before eating or feeding the child | 0.89 | 0.92 | 0.37 | – | – |
| **Domain: family planning** | | | | | | |
| How soon after you give birth can you get pregnant again? | Immediately | 0.02 | 0.01 | 0.04 | 0.43 | −0.02 |
| | Not until menses return | 0.73 | 0.8 | 0.12 | – | – |
| | After you stop breast feeding | 0.03 | 0.01 | 0.14 | – | – |
| What is the recommended length of time you should wait between having another child? | Immediately | 0.00 | 0.00 | 0.00 | 0.61 | 0.36 |
| | Wait for at least 1 year | 0.02 | 0.02 | 0.02 | – | – |
| | Wait for at least 2 years | 0.19 | 0.27 | 0.18 | – | – |
| | Wait for at least 3 years | 0.67 | 0.64 | 0.51 | – | – |
| What are the benefits of using family planning to limit the size of your family? | Easy way to control the size of your family | 0.48 | 0.47 | 0.46 | 0.02 | 0.00 |
| | Financial savings | 0.50 | 0.51 | 0.50 | 0.01 | 0.06 |
| | Give you more time to take care of the children you already have | 0.70 | 0.82 | 0.54 | – | – |
| Men become physically weak after accepting sterilisation | False | 0.15 | 0.14 | 0.09 | 0.28 | −0.02 |
| There are many safe methods of birth control | True | 0.94 | 0.96 | 0.33 | – | – |
| Female sterilisation can be done at the time of birth | True | 0.76 | 0.82 | 0.21 | – | – |
| Male sterilisation is an easy way to control family size | True | 0.72 | 0.63 | 0.22 | – | – |
| Postpartum intrauterine device insertion and female sterilisation | True | 0.87 | 0.86 | 0.35 | – | – |
| Do you know of a place where you can obtain a method of family planning? | Yes | 0.9 | 0.98 | 0.38 | – | – |
| Which ways or methods of contraception have you heard about? | Female sterilisation | 0.92 | 0.97 | 0.37 | – | – |
| | Male Sterilisation | 0.49 | 0.54 | 0.21 | – | – |
| | IUD | 0.47 | 0.36 | 0.31 | – | – |
| | Oral contraceptives | 0.84 | 0.93 | 0.58 | – | – |
| | Injectables | 0.70 | 0.78 | 0.28 | – | – |
| | Male condom | 0.64 | 0.74 | 0.46 | – | – |
| | Rhythm method | 0.56 | 0.45 | 0.02 | – | – |
| | Withdrawal | 0.24 | 0.08 | 0.05 | – | – |
| **Domain: essential newborn care** | | | | | | |

**Table 2** Continued

| Question text | Response options | Baseline prevalence | Retest prevalence | Phone prevalence | Retest kappa | Phone kappa |
|---|---|---|---|---|---|---|
| How do you keep the baby warm after delivery? | Put the baby on your chest | 0.17 | 0.15 | 0.13 | 0.5 | 0.49 |
| | Dried or wiped soon after birth | 0.25 | 0.30 | 0.24 | 0.08 | 0.24 |
| | Cover in clothes | 0.96 | 1.00 | 0.56 | | |
| What should you put on the cord after delivery? | Nothing | 0.06 | 0.05 | 0.04 | 0.82 | 0.77 |
| | Blade used for other purposes | 0.72 | 0.72 | 0.05 | – | – |
| | Scissors | 0.41 | 0.72 | 0.29 | – | – |
| | Knife | 0.02 | 0.03 | 0.18 | – | – |
| | Surgical blade | 0.03 | 0.01 | 0.39 | – | – |
| | New razor blades | 0.72 | 0.71 | 0.11 | – | – |
| How soon after delivery should your baby be bathed? | 1 day | 0.8 | 0.77 | 0.03 | 0.49 | 0.8 |

IFA, iron folic acid.

Honduras and Peru compared additional phone survey modalities, including text messages and interactive voice response (IVR), and concluded that CATI surveys had the lowest discordance with the face-to-face surveys.[24] Elsewhere, more recent efforts have sought to compare the reliability of alternative phone survey delivery modalities (IVR vs CATI) for measuring non-communicable disease risk factors.[28] In our context, low literacy rates precluded testing phone survey modalities other than CATI surveys.

Underpinning efforts to rigorously develop the phone survey tool was a design to measure knowledge among pregnant women. Overall knowledge levels among pregnant women in these four districts of Madhya Pradesh ranged from 47% for essential newborn care to 58% for family planning. Areas of lower knowledge included what should be put on the cord after delivery, how to keep the baby warm after delivery, what foods to eat during pregnancy, what food and drinks to avoid during pregnancy, etc. Knowledge scores were significantly higher among older women; those with higher levels of education and literacy; living in a nuclear family; who reported being the primary decision-maker in health decisions; who had attended a greater number of ANC visits and reported being satisfied with ASHA services.

Other studies have measured women's knowledge levels on topics such as neonatal danger signs, immediate post-birth care for the neonate, infant and young child feeding, breast feeding and family planning. Compared with 18% of women in Ethiopia and 15% of women in Uganda who have good knowledge of neonatal danger signs, our women have a higher relative knowledge at 58%.[29 30] In a survey among hospital-delivered mothers in Sri Lanka, while 90% of mothers knew about advantages of breast feeding compared with our survey of 56%, only 22% correctly answered what should be applied to an umbilical cord compared with our survey at 47%.[31] In India, universal knowledge existed among married women in Uttar Pradesh about at least one family planning method and 90% of women in Lucknow were aware of contraceptive methods in family planning existed comparatively to our study's lower family planning knowledge at 58%.[32 33] This is all subject to each study's own verified survey methods and composite score generation. Developing one standardised survey questionnaire across

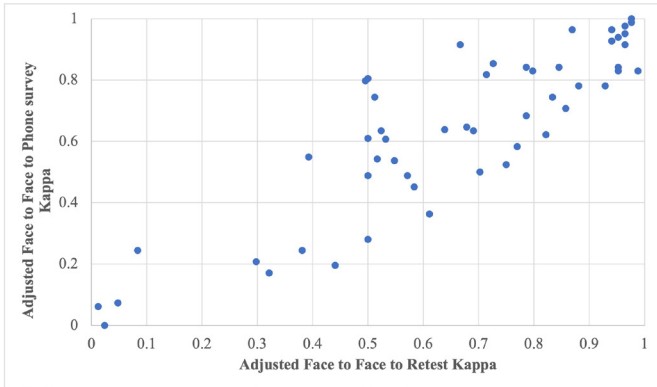

**Figure 3** Measurement of degree in which repeated measurements in pregnant women interviewed (test–retest) provide similar answers.

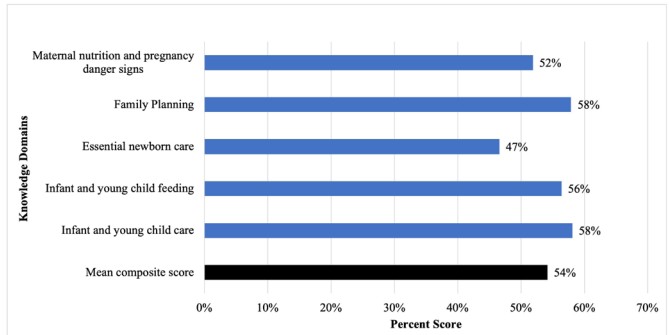

**Figure 4** Average knowledge scores for composite knowledge domains.

**Table 3** Factors associated with pregnant women's RMNCH&N knowledge in four districts of Madhya Pradesh, India

| Variable | Unadjusted coefficient (95% CI) | P value | Adjusted coefficient (95% CI) | P value |
|---|---|---|---|---|
| **Age** | | | | |
| 18–24 | 1 | — | 1 | — |
| 25–34 | 3.21 (2.68 to 3.74) | <0.001 | 1.91 (1.35 to 2.46) | <0.001 |
| 35+ | 1.87 (−0.04 to 3.78) | 0.055 | 2.04 (0.34 to 3.74) | 0.02 |
| **Parity (≥1 child)** | 4.40 (3.85 to 4.96) | <0.001 | 4.39 (3.74 to 5.03) | <0.001 |
| **Caste** | | | | |
| General | 1 | — | 1 | — |
| Other Backward Caste | −0.62 (−1.30 to 0.05) | 0.070 | 0.62 (−0.06 to 1.3) | 0.076 |
| Scheduled Caste/ Tribe | −0.97 (−1.70 to 0.24) | 0.010 | 0.24 (−0.58 to 1.06) | 0.567 |
| **Wealth index** | | | | |
| Q1 | 1 | — | 1 | — |
| Q2 | 0.30 (−0.52 to 1.13) | 0.471 | 0.32 (−0.49 to 1.13) | 0.442 |
| Q3 | 0.03 (−0.80 to 0.86) | 0.938 | 0.14 (−0.78 to 1.07) | 0.765 |
| Q4 | 0.25 (−0.58 to 1.08) | 0.551 | 0.16 (−0.83 to 1.16) | 0.751 |
| Q5 | 1.83 (1.00 to 2.66) | <0.001 | 0.72 (−0.38 to 1.82) | 0.197 |
| **Education level** | | | | |
| No school | 1 | — | 1 | — |
| Primary or less | 0.71 (−0.31 to 1.72) | 0.172 | 0.65 (−0.35 to 1.65) | 0.201 |
| Middle school | 1.32 (0.39 to 2.24) | 0.005 | 0.6 (−0.51 to 1.72) | 0.289 |
| Higher education | 3.26 (2.37 to 4.16) | <0.001 | 1.65 (0.42 to 2.89) | 0.009 |
| **Literacy** | | | | |
| Cannot read at all | 1 | — | 1 | — |
| Reading only parts of sentence | 1.43 (0.52 to 2.34) | 0.002 | 0.95 (0.01 to 1.9) | 0.049 |
| Read whole sentence | 3.02 (2.44 to 3.59) | <0.001 | 2.29 (1.42 to 3.15) | <0.001 |
| **Self-help group** | 0.97 (−0.03 to 1.97) | 0.057 | 0.4 (-0.42 to 1.22) | 0.335 |
| **Nuclear family structure** | −1.97 (−2.61 to −1.33) | <0.001 | −2.14 (−2.95 to −1.33) | <0.001 |
| **Women is the primary decision-maker in health decisions** | 2.03 (1.45 to 2.61) | <0.001 | 1.59 (1 to 2.19) | <0.001 |
| **Number of ANC visits** | | | | |
| 1 | 1 | — | 1 | — |
| 2 | 1.13 (0.27 to 2.01) | 0.011 | 1.03 (0.16 to 1.9) | 0.021 |
| 3 | 1.64 (0.73 to 2.56) | <0.001 | 1.95 (1.04 to 2.85) | <0.001 |
| 4 | 1.14 (0.34 to 1.94) | 0.005 | 1.68 (0.81 to 2.54) | <0.001 |
| **Satisfied with the services provided by the ASHA** | 2.61 (2.02 to 3.19) | <0.001 | 2.18 (1.57 to 2.79) | <0.001 |
| **Phone ownership** | 1.31 (0.71 to 1.93) | <0.001 | 0.61 (−0.01 to 1.22) | 0.055 |
| **District** | | | | |
| Hoshangabad | 1 | — | 1 | — |
| Mandsaur | −3.30 (−4.42 to −2.17) | <0.001 | −3.3 (−4.51 to −2.09) | <0.001 |
| Rewa | −1.53 (−2.52 to −0.55) | <0.001 | −0.13 (−1.21 to 0.95) | 0.814 |

Continued

**Table 3** Continued

| Variable | Unadjusted coefficient (95% CI) | P value | Adjusted coefficient (95% CI) | P value |
|---|---|---|---|---|
| Rajgarh | −4.65 (−5.75 to −3.55) | <0.001 | −3.05 (−4.21 to −1.89) | <0.001 |

ANC, antenatal care; ASHA, accredited social health activist; RMNCH&N, reproductive, maternal, newborn, child health and nutrition.

studies to measure women's RMNCH&N knowledge would lend more insight into how comparative these data are.

There is limited empirical evidence of factors influencing pregnant women's knowledge. A study conducted in Delhi assessed knowledge components that included immunisation against tetanus, number of antenatal visits, folic acid tablets, danger signs during pregnancy or labour, and discussions with elders, mothers-in-law, doctors, or friends.[34] Like our study, it found that women in joint families were also significantly more likely to have higher knowledge.[34] Another study in Rewa district also found that women whose husbands and in-laws played a dominant role in decision-making had lower knowledge of key danger signs during pregnancy.[35] This is consistent with our study, which included Rewa district among our four study districts, as we found if the women played a role in the decision-making process, she had significantly higher knowledge of maternal danger signs. In a Uganda study, no association was found with attending ANC visits and knowledge of danger signs, which is inconsistent with our findings.[30] Surprisingly, while NFHS found that more family planning methods were seen in women of higher wealth index, wealth was not found to be a determinant in this study.[11] Taken together, our findings suggest that to increase women's knowledge in the future, one should empower women to play a larger role in the health decision-making process, strengthen ASHA services, promote education, and increase access and uptake of ANC services. Further research to understand sources and quality of sources for knowledge should be undertaken as well.

## Limitations

This survey was limited to RMNCH&N knowledge in women 5–7 months pregnant with access to a mobile phone during the day, in four predominately rural districts of Madhya Pradesh. This may have led to a pro-rich bias in the sampling given the profile of beneficiaries with access to mobile phones.[13] Survey questions were derived from expert review and large national survey tools, including the demographic and health survey for India and the Multiple Indicator Cluster Survey. Cognitive interviews were used to refine questions on respectful maternity care, a process described elsewhere.[19 20 36] Ideally, the same process would have been followed for refining knowledge questions. However, budget and time constraints made this added step infeasible. Phone surveys were administered by Delhi-based enumerators whose demographic profile differed from respondents.

Alternative phone survey modalities, including text messages, IVR and Unstructured Supplementary Service Data (USSD), were considered but deemed less desirable to using CATI surveys, which could allow beneficiaries to speak with and ask for clarification questions.

## CONCLUSIONS

Study findings offer insight into the challenges associated with reliably measuring RMNCH&N knowledge among pregnant women using face-to-face and remote survey tools. Population-level estimates of knowledge over time and across modalities were similar, but reliability among individual women in our sample was poor. This suggests that the use of phone surveys to measure knowledge is contraindicated in instances where longitudinal tracking or individual-level analyses are required; however, it may be appropriate as a means of gaining population-level insights. Among women assessed, overall knowledge levels were low; however, knowledge scores were significantly higher among older women; those with higher levels of education and literacy; living in a nuclear family; who reported being the primary decision-maker in health decisions; who had attended a greater number of ANC visits and reported being satisfied with ASHA services.

**Author affiliations**
[1]Department of International Health, Johns Hopkins Bloomberg School of Public Health, Baltimore, Maryland, USA
[2]Oxford Policy Management, New Delhi, Delhi, India
[3]BBC Media Action, New Delhi, Delhi, India
[4]BBC Media Action, London, UK
[5]Health, Nutrition and Population, World Bank New Delhi Office, New Delhi, India
[6]The Bill and Melinda Gates Foundation, Seattle, Washington, USA
[7]National Health Systems Resource Centre, New Delhi, Delhi, India
[8]The Bill and Melinda Gates Foundation, Delhi, India
[9]Division of Public Health and Family Medicine, University of Cape Town, School of Public Health and Family Medicine, Cape Town, South Africa

**Acknowledgements** We thank the women and families of Madhya Pradesh who generously gave their time to support this work. We are humbled by the opportunity to convey their perspectives and experiences. We additionally are grateful to Neha Dumke and the team at the National Health Systems Resource Centre (NHSRC) for their support. This work was made possible by the Bill and Melinda Gates Foundation. We thank Diva Dhar, Suhel Bidani, Rahul Mullick, Dr Suneeta Krishnan, Dr Neeta Goel and Dr Priya Nanda for believing in us and giving us this opportunity. We additionally wish to thank the larger team of enumerators from OPM-India and NHSRC who worked tirelessly over many months to implement the surveys that form the backbone of our analyses. We additionally thank Vinit Pattnaik at OPM and Erica Crawford at Johns Hopkins University for the financial management support of our work.

**Collaborators** Kilkari Impact Evaluation Team: Smisha Agarwal, Salil Arora, Jean JH Bashingwa, Aarushi Bhatanagar, Sara Chamberlain, Rakesh Chandra, Arpita Chakraborty, Neha Dumke, Priyanka Dutt, Anna Godfrey, Suresh Gopalakrishnan, Nayan Kumar, Simone Honikman, Alain Labrique, Amnesty LeFevre, Jai Menditratta, Molly Miller, Radharani Mitra, Diwakar Mohan, Deshen Moodley, Nicola Mulder, Angela Ng, Dilip Parida, Nehru Penugonda, Sai Rahul, Shiv Rajput, Neha Shah, Kerry Scott, Aashaka Shinde, Aaditya Singh, Nicki Tiffin, Osama Ummer, Rajani Ved, Falyn Weiss, Sonia Whitehead.

**Contributors** AEL is the overall study PI and conceived the idea for this study along with DM, AB, RV, KS and SC. AEL is responsibile for the overall content and serves as the guarantor. Data collection tools were designed by AEL, NS, DM, KS, AB, SA, OU and RV with inputs from DD. Phone survey data collection was led and supervised by NS and RV. Face-to-face data collection was supervised by AB, OU, AEL, DM and AEL. AN and NS led the analyses with oversight from DM and inputs from AEL. AN and AEL wrote the manuscripts with edits and inputs from all coauthors. AN generated all final tables and figures.

**Funding** Bill and Melinda Gates Foundation.

**Disclaimer** The study sponsors had no role in the design, implementation or interpretation of study findings. The study team were independent from the funders.

**Competing interests** All authors have completed the Unified Competing Interest form (available on request from the corresponding author) and declare that the research reported was funded by the Bill and Melinda Gates Foundation. AG, SC, PD are employed by BBC Media Action, one of the entities supporting programme implementation. The authors do not have other relationships and are not engaged in activities that could appear to have influenced the submitted work.

**Patient consent for publication** Not required.

**Ethics approval** Institutional Review Boards from the Johns Hopkins Bloomberg School of Public Health in Baltimore, Maryland, USA and Sigma Research and Consulting in Delhi, India provided ethical clearance for study activities. Verbal informed consent was obtained from all study participants.

**Provenance and peer review** Not commissioned; externally peer reviewed.

**Data availability statement** Extra data can be found by emailing Amnesty LeFevre at aelefevre@gmail.com Not applicable.

**ORCID iDs**
Angela Ng http://orcid.org/0000-0002-1872-6665
Diwakar Mohan http://orcid.org/0000-0002-7532-366X
Neha Shah http://orcid.org/0000-0002-9450-604X
Kerry Scott http://orcid.org/0000-0003-3597-9637
Osama Ummer http://orcid.org/0000-0002-4189-5328
Sara Chamberlain http://orcid.org/0000-0003-4785-6482
Amnesty Elizabeth LeFevre http://orcid.org/0000-0001-8437-7240

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
