## [Reviewer comments · BMJ Open]

ARTICLE DETAILS

TITLE (PROVISIONAL)	Assessing the reliability of phone surveys to measure reproductive, maternal, and child health knowledge among pregnant women in rural India: a feasibility study
AUTHORS	Ng, Angela; Mohan, Diwakar; Shah, Neha; Scott, Kerry; Ummer, Osama; Chamberlain, Sara; Bhatnagar, Aarushi; Dhar, Diva; Agarwal, Smisha; Ved, Rajani; LeFevre, Amnesty

VERSION 1 – REVIEW

REVIEWER	Wayessa, Zelalem Bule Hora University, Midwifery
REVIEW RETURNED	23-Aug-2021

GENERAL COMMENTS	Title: - Optimizing the measurement of reproductive, maternal, and newborn health and nutrition knowledge among pregnant women in India: are phone surveys a reliable modality for data collection? With "...only 29% of women report having access to mobile phones that they can use" women, how did you overcome this study? And how are phone surveys a reliable modality for data collection? Abstract - Under the method section: - Study period should be included.- Under the result section: - The total study participants with their response rate should be included- Keywords should be included Introduction - The authors shall rewrite it clearly showing what is known, what is unknown, what they were interested in to deal with this problem as a summary. Even the discussed one was an outdated figure (please briefly write the actual problems with the evidence) Eg. Despite declines in maternal and child mortality from 1990 to 2015 of 77% and 60% respectively, India fell short of attaining the millennium develop goals for maternal and child health (1). What about 2016- 2021? Methods Study setting. The statements "...Significant gaps between men and women persist with regard to literacy (82% of men are literate as compared to 59% of women), and only 29% of women report having access to mobile phones that they can use (10). For nearly all health indicators, Madhya Pradesh falls below national averages. Despite near universal attendance of at least one antenatal care visit during pregnancy, only 53% of women receive services within the first trimester, and 36% receive the recommended four visits (10). While institutional delivery rates have increased over time, 20% of women deliver outside the formal health sector, and only 18% receive a health check after
---

	birth from a trained provider within 2 days of birth (10). Among children, 51 per 1,000 live births will die prior to their first birthday and 65 per 1,000 live births will die prior to their fifth birthday (10). High mortality rates are underpinned by undernutrition (43% of children are under weight, 42% are stunted), low rates of exclusive breastfeeding (58%), and gaps in access to basic health services including immunization. Where the latter is concerned, only 73% of children 12-23 months have received the recommended 3 doses of the DPT vaccine (10)" are the statement of the problem. So it's better if you included it under the introduction section. Study design and sampling  • Who is your target population?  [ ] Women with reported access to a mobile phone might include women out of reproductive age groups. You aimed to illuminate gaps in RMNCH&N knowledge among pregnant women.... what is your justification? • The sample size calculation was not clear, it needs clarification. • What is your sampling technique? Data collection  • What is the evidence if somebody responses to your call for a phone survey? How did you control information bias? • What do you think the reliability and validity of knowledge questions responses by phone surveys? • The validity of your measurements tools is not mentioned? Are that standardized tools? "Average knowledge scores for the five composite knowledge domains....." before you merge all five domains, do you have to operationalize every five domains? Results  - How many study participants were included for each face-to-face survey and phone survey with their response rate? It should be mentioned - What are cut points to say knowledgable or not for all domains? - Where is the specific result obtained from the phone survey? Discussion  - Your result section and discussion section are similar, I haven't seen any other findings? So what makes it different? - The authors shall scientifically justify their findings in a specific and convincing way Conclusions  - The conclusion should be based on your finding. The result is talking about face-to-face surveys, but your conclusion was talking about phone survey results. - It needs revision Note: - The title seems innovative but the work needs extensive revisions
--	--

REVIEWER	Sialubanje, Cephas Ministry of Health, Levy Mwanawasa Medical University
REVIEW RETURNED	31-Aug-2021

GENERAL COMMENTS	This is a survey nested in an impact evaluation of a mobile health intervention study conducted among 5,095 women with reported access to a mobile phone during the day in four districts of the Central stat of India. The aim of the study was assess differences in the reliability of the knowledge questions over different modalities including face-to-face surveys at two time points and caller attended telephone interviews. Overall, the study provides important information on how to measure pregnant women's knowledge.
--

	Nevertheless, authors can enrich their study by attending to the following points: Minor comments:  • Page 7 lines 4-5....in addition to the percentage reduction, let the authors indicate the current maternal mortality ratio and child mortality rate • Page 7 line 32-34 citation is missing; please include • Page 7 line 41 should read predominantly instead of predominates • Page 8 line 3-4 please describe in brief the impact evaluation and that this study was nested in it; this will make it easier for the reader • Page 8: 27-28...what does general pregnancy mean? Please clarify for the reader • Page 8:36-37 should read comprised factors, and not comprised of factor; consider replacing the word "underpin" with influence or affect; do so everywhere it appears in the manuscript • Page 9 line: should read ethical approval and not ethics approval • Page 9 line 25 should read other castes, and not other backward castes; backward sounds demeaning • Page 10 Table 2: indicate what factors were adjusted for • Page 12 line 36....should read the aim of this study.... Discussion  • In general, the discussion is quite weak. • Results are not discussed nor corroborated with previous studies • It is merely a repetition of methods and results • An attempt to corroborate the results is made in the third paragraph (page 13 lines 34 to 53, but not quite well done Let the authors recast the discussion:  • First paragraph should state the aim and summarise the major findings • Subsequent paragraphs should discuss the results in light of what is known, as well as their contribution to the body of knowledge in this scientific field of study Conclusion:  • The conclusion is poorly presented; let the authors: • Summarise the major findings • Clearly state the implications for science and practice • Clearly formulate recommendations Major corrections  • Page 12 line 3-33 it is not clear which measure was used for inter-rater reliability and intermodal reliability. Elsewhere the authors mention about Kappa and Chronibach's alpha; but only kappa results are reported here
--	---

REVIEWER	Nair, Divya ID Insight Private Limited
REVIEW RETURNED	10-Sep-2021

GENERAL COMMENTS	The study looks at important questions around knowledge of RMNCH and its measurement in Madhya Pradesh. It sheds light on the types of questions that are more reliably measured, and it is innovative and timely in layering on an assessment of how phone surveys can be leveraged to measure these dimensions. This is particularly relevant given the pandemic and the pivot towards phone surveys. It would be useful to have more clarity on the following:
---

	1. The objective of this paper can be sharpened. Do the authors want to shed light on the levels of/ reasons for low knowledge of key RMNCH topics, or do they want to delve into the pros and cons of various measurement techniques to assess this knowledge. The paper would be stronger if the authors choose one of these paths and then delve deeper. For example, if the goal is to shed light on lack of knowledge on RMNCH topics, it would be useful to know more about the context of Madhya Pradesh and include information on the supply side – what are the sources of knowledge for these respondents; how good is the quality of those sources, discuss district variation, etc. If the goal is to create a new tool and examine different measurement techniques, the authors should provide more information about the framing of questions; training protocols; how these protocols transferred across modes (e.g did they have female phone surveyors). In either case, the paper should also discuss how the context mattered, and how transferrable these results are, and to whom. 2. Who the participants in the study are/ who they represent is unclear (a) It would be useful to describe in more detail the sampling strategy and selection criteria of respondents. [Referring to another article for the protocol is not sufficient] (b) It appears that a majority of the respondents ended up being from Rewa district in Madhya Pradesh; how representative is this population of other rural/Indian populations especially if the focus of this article is on knowledge levels? [Triangulating with NFHS or other sources could help to understand who this sample represents] (c) If the focus is on reliability across different modes, what was the nature of attrition/refusal across the different types of surveys (i.e. were the women who consented to the phone surveys different from those who did not)? 3. Additional questions:  • The authors refer interchangeably to test-retest reliability and inter-rater reliability, they should clarify this. • The authors do not address question of validity. Did the respondents understand the questions well/better in a particular mode; which mode do the authors believe reflects respondents' knowledge better? • What sort of training / protocols were followed by surveyors - were the surveyors the same, does the study account for surveyor fixed effects? • The authors hypothesize on line 20 (page 13) that the “reasons for low reliability could include recall bias (especially for open-ended questions with multiple coded response options), social desirability, perceived redundancies in answer options, and enumerator dependent ambiguous answers”. This sentence is unclear. If the response is different from earlier this could be because (a) for a given respondent sample, knowledge has changed; understanding of the question was unclear at the earlier time point; understanding of the question is unclear now; or the respondents are just guessing. But, (b) it could also be that the respondent sample has changed. It would be useful to clarify what the authors think may be happening.
--	---

VERSION 1 – AUTHOR RESPONSE

	Reviewer comment	Author responses
1	- Please revise your title so that it includes your study design. This is the preferred format for the journal.	Thank you. We have revised the title to: “Assessing the reliability of phone surveys to measure reproductive, maternal, and child health knowledge among pregnant women in rural India: a feasibility study”
2	- Please revise the abstract so that it is following the structured abstract recommended in BMJ Open's Instructions for Authors for research articles. See: https://bmjopen.bmj.com/pages/authors/#research	The abstract is revised and follows BMJ Open's designated structure
3	- Please add the relevant numbers and statistics to support your statements in the results section of the abstract.	Thank you. The relevant numbers and statistics were added to help provide context clues to the results section of the abstract.
4	- Please revise the strengths and limitations section after the abstract. It should contain up to five short bullet points, no longer than one sentence each, that relate specifically to the methods of the study reported (see: http://bmjopen.bmj.com/site/about/guidelines.xhtml#articletypes). It should not be a summary of the study and its findings.	Thank you. We have amended this section to read as follows: Strengths:  - COVID-19 has accelerated the use of remote data capture strategies, however, few studies have undertaken a rigorous process of tool development inclusive of reliability testing. - Study findings suggest that phone surveys can be a reliable modality for measuring knowledge among pregnant women in rural India at a population level. - To optimize reliability, limit the use of questions which are sensitive, or have open-ended or multi-response options. Limitations: The sample included pregnant women with reported access to a phone during the day; a population arguably more advantaged than women without access to a phone

		which could limit the generalizability of findings.
5	- Can you please work on improving the patient and public involvement section? Some of the information provided here does not seem to relate to PPI. You also do not need to include the questions in this section. The Patient and Public Involvement statement should be included as a sub-heading in the methods section of all manuscripts. It should provide a brief description of any patient involvement in study design or conduct of the study, as well as any plans to disseminate the results to study participants. If patients and or public were not involved then please state this. The Patient and Public Involvement statement should not contain details of participant recruitment, patient consent or ethics approval. This information should be included elsewhere in your methods section. Please see our blog for further information regarding PPI: http://blogs.bmj.com/bmjopen/2018/03/23/new-requirements-for-patient-and-public-involvement-statements-in-bmj-open/	Thank you. This was moved to the methods section of our manuscript.
6	With "...only 29% of women report having access to mobile phones that they can use" women, how did you overcome this study? And how are phone surveys a reliable modality for data collection?	There is no means to overcome population level limitations in phone access without specific initiatives to bolster the complex determinants underpinning phone access and use. We explore these concepts in other papers in the Special Supplement, which we hope this paper will be published alongside. In the discussion and limitations sections of this paper we highlight the limits of phone surveys in this population and note that reliability findings are only generalisable to a sample with access to a phone. Amongst this population with access to a phone, phone surveys are a reliable modality for knowledge measurement at a population level.

7	- Under the method section: - Study period should be included.	Thank you for catching this. We have amended the methods section to note that the study period was from August 2018-October 2018.
8	- Under the result section: - The total study participants with their response rate should be included	Thank you. We have amended the results.
9	- Keywords should be included	Thank you. We have added “phone surveys, knowledge, India and pregnant women”
10	- The authors shall rewrite it clearly showing what is known, what is unknown, what they were interested in to deal with this problem as a summary. Even the discussed one was an outdated figure (please briefly write the actual problems with the evidence) Eg. Despite declines in maternal and child mortality from 1990 to 2015 of 77% and 60% respectively, India fell short of attaining the millennium develop goals for maternal and child health (1). What about 2016- 2021?	We have added new citations with the latest available data - 2018 for maternal mortality, and 2019 for child mortality.
11	Study setting. The statements “....Significant gaps between men and women persist with regard to literacy (82% of men are literate as compared to 59% of women), and only 29% of women report having access to mobile phones that they can use (10). For nearly all health indicators, Madhya Pradesh falls below national averages. Despite near universal attendance of at least one antenatal care visit during pregnancy, only 53% of women receive services within the first trimester, and 36% receive the recommended four visits (10). While institutional delivery rates have increased over time, 20% of women deliver outside the formal health sector, and only 18% receive a health check after birth from a trained provider within 2 days of birth (10). Among children, 51 per 1,000 live births will die prior to their first birthday and 65 per 1,000 live births will die prior to their fifth birthday (10). High mortality rates are underpinned by undernutrition (43% of children are under weight, 42% are stunted), low rates of exclusive breastfeeding (58%), and gaps in access to basic health services including immunization. Where the latter is concerned,	Thank you. We respectfully wish to retain this under methods. To move it to the introduction would shift focus away from our central problem statement around limitations in the ways in which phone survey tools have been developed. The broader health burden is included under settings for context.

	only 73% of children 12-23 months have received the recommended 3 doses of the DPT vaccine (10)” are the statement of the problem. So it’s better if you included it under the introduction section.	
12	Study design and sampling  • Who is your target population?) Women with reported access to a mobile phone might include women out of reproductive age groups. You aimed to illuminate gaps in RMNCH&N knowledge among pregnant women.... what is your justification? • The sample size calculation was not clear, it needs clarification. • What is your sampling technique? 	Thank you. We have added the requested details to the methods section, including details on the sample size calculation and sampling technique. We note that data were collected as part of a larger impact evaluation. Hence the sample is restricted to women 5-7 months pregnant with access to a phone.
13	Data collection  • What is the evidence if somebody responses to your call for a phone survey? How did you control information bias? • What do you think the reliability and validity of knowledge questions responses by phone surveys? • The validity of your measurements tools is not mentioned? Are that standardized tools? “Average knowledge scores for the five composite knowledge domains.....” before you merge all five domains, do you have to operationalize every five domains? 	The primary focus of our analysis is on survey tool content; hence the analytic focus on completed surveys which enable 1:1 comparison of responses to questions by modality. Intermodal reliability results are presented in the results and the text has been amended to better flesh out findings. The origins of surveys tools are described in-depth elsewhere. In brief, questions were drawn from standardised national survey tools, the literature and expert judgment and enhanced through cognitive testing and piloting. We have amended the methods to include these details.
14	Results  - How many study participants were included for each face-to-face survey and phone survey with their response rate? It should be mentioned - What are cut points to say knowledgable or not for all domains? - Where is the specific result obtained from the phone survey? 	Thank you. The number of study participants for test-retest/phone survey were added in the results in the survey reliability portion. Cut points were not used to ascertain knowledge. Text in the methods describe our approach for generating mean composite scores.
15	Discussion  - Your result section and discussion section are similar, I haven’t seen any other findings? So what makes it different? 	Thank you. The discussion section was updated to be different then the results section.

	 - The authors shall scientifically justify their findings in a specific and convincing way 	
16	Conclusions  - The conclusion should be based on your finding. The result is talking about face-to-face surveys, but your conclusion was talking about phone survey results. - It needs revision 	Thank you. The conclusions were updated to add findings about the phone survey results.
17	Minor comments:  • Page 7 lines 4-5....in addition to the percentage reduction, let the authors indicate the current maternal mortality ratio and child mortality rate • Page 7 line 32-34 citation is missing; please include • Page 7 line 41 should read predominantly instead of predominates • Page 8 line 3-4 please describe in brief the impact evaluation and that this study was nested in it; this will make it easier for the reader • Page 8: 27-28...what does general pregnancy mean? Please clarify for the reader • Page 8:36-37 should read comprised factors, and not comprised of factor; consider replacing the word "underpin" with influence or affect; do so everywhere it appears in the manuscript • Page 9 line: should read ethical approval and not ethics approval • Page 9 line 25 should read other castes, and not other backward castes; backward sounds demeaning • Page 10 Table 2: indicate what factors were adjusted for • Page 12 line 36....should read the aim of this study.... 	Thank you, we have made the adjustments in response to these comments. The Kappa coefficients were adjusted for the difference in prevalence levels using the Prevalence Adjusted Bias Adjusted Kappa (PABAK) score and listed in the Data Analysis, but we have added it again right before introducing Table 2. Other Backward Class (OBC) is a collective term used by the Government of India to classify castes which are educationally or socially disadvantaged. It is one of several official classifications of the population of India, along with General Class, Scheduled Castes and Scheduled Tribes. We have chosen to keep the word backward for this reason.
18	Discussion  • In general, the discussion is quite weak. • Results are not discussed nor corroborated with previous studies • It is merely a repetition of methods and results • An attempt to corroborate the results is made in the third paragraph (page 13 lines 34 to 53, but not quite well done 	Thank you. The discussion has been updated to restate the aim and summarise major findings, and subsequent paragraphs discuss the results in light of what is known and their contribution to the future of mobile surveys and pregnant women's knowledge.

	Let the authors recast the discussion:  • First paragraph should state the aim and summarise the major findings • Subsequent paragraphs should discuss the results in light of what is known, as well as their contribution to the body of knowledge in this scientific field of study 	
19	Conclusion:  • The conclusion is poorly presented; let the authors: • Summarise the major findings • Clearly state the implications for science and practice • Clearly formulate recommendations 	We have revised the conclusion to summarise the major findings
20	 • Page 12 line 3-33 it is not clear which measure was used for inter-rater reliability and intermodal reliability. Elsewhere the authors mention about Kappa and Chronibach's alpha; but only kappa results are reported here 	Kappa was used to measure both inter-rater reliability and intermodal reliability and has been presented here. Cronbach's Alpha of 0.7 is used as a cutoff for the kappa scores.
21	1. The objective of this paper can be sharpened. Do the authors want to shed light on the levels of/ reasons for low knowledge of key RMNCH topics, or do they want to delve into the pros and cons of various measurement techniques to assess this knowledge. The paper would be stronger if the authors choose one of these paths and then delve deeper. For example, if the goal is to shed light on lack of knowledge on RMNCH topics, it would be useful to know more about the context of Madhya Pradesh and include information on the supply side – what are the sources of knowledge for these respondents; how good is the quality of those sources, discuss district variation, etc. If the goal is to create a new tool and examine different measurement techniques, the authors should provide more information about the framing of questions; training protocols; how these protocols transferred across modes (e.g did they have female phone surveyors). In either case, the paper should also discuss how the context mattered, and how transferrable these results are, and to whom.	Thank you for this insightful comment. Our principal focus is on modalities for measuring outcomes -- knowledge in this case. We have sought to improve clarity of focus by re-shaping the introduction to focus more on phone surveys and methods for developing phone survey tools. We hope that this addresses your very valid concerns.

22	Who the participants in the study are/ who they represent is unclear (a) It would be useful to describe in more detail the sampling strategy and selection criteria of respondents. [Referring to another article for the protocol is not sufficient] (b) It appears that a majority of the respondents ended up being from Rewa district in Madhya Pradesh; how representative is this population of other rural/Indian populations especially if the focus of this article is on knowledge levels? [Triangulating with NFHS or other sources could help to understand who this sample represents] (c) If the focus is on reliability across different modes, what was the nature of attrition/refusal across the different types of surveys (i.e. were the women who consented to the phone surveys different from those who did not)?	a) The sampling strategy and selection criteria of respondents has been added in this paper b) The population of Rewa is representative of the population living in the Gangetic plain. c) The number of women who refused a follow up survey was very small so it was intangible to calculate the nature of attrition
22	 The authors refer interchangeably to test-retest reliability and inter-rater reliability, they should clarify this. 	Thank you, this has been revised in the paper; we referred to the survey as test-retest and reliability as inter-rater reliability.
23	 The authors do not address question of validity. Did the respondents understand the questions well/better in a particular mode; which mode do the authors believe reflects respondents' knowledge better? 	The face-to-face mode was understanding baseline knowledge, with the goal to see if phone surveys could reproduce that knowledge, not necessarily to see if phone surveys were better than face-to-face surveys.
24	 What sort of training / protocols were followed by surveyors - were the surveyors the same, does the study account for surveyor fixed effects? 	Thank you, we added this in the Methods section in Data Collection. Surveyors were all graduate students who went through a training that involved reviewing the interview form and answering any questions they may have. They were expected to call participants and complete surveys in one setting or schedule interviews for a later time. Because surveyors were overseen daily, any issues or questions were clarified in real time and any concerns or deviations were addressed as soon as possible.
25	 The authors hypothesize on line 20 (page 13) that the "reasons for low reliability could include recall bias (especially for open-ended questions with multiple coded response options), social desirability, perceived redundancies in answer options, and enumerator dependent 	Thank you for these insights, we have expanded upon this comment in the discussion portion of our paper. As the test-retest and phone surveys were repeated just after two weeks, it was not expected that they had any intervention change for knowledge, but rather the questions were not

	ambiguous answers". This sentence is unclear. If the response is different from earlier this could be because (a) for a given respondent sample, knowledge has changed; understanding of the question was unclear at the earlier time point; understanding of the question is unclear now; or the respondents are just guessing. But, (b) it could also be that the respondent sample has changed. It would be useful to clarify what the authors think may be happening.	designed well or able to be repeated over the phone.
--	---	--

VERSION 2 – REVIEW

REVIEWER	Sialubanje, Cephas Ministry of Health, Levy Mwanawasa Medical University
REVIEW RETURNED	06-Dec-2021

GENERAL COMMENTS	The authors have addressed the earlier comments. The manuscript can be considered for publication
---

REVIEWER	Nair, Divya ID Insight Private Limited
REVIEW RETURNED	07-Feb-2022

GENERAL COMMENTS	1. The manuscript needs close proofreading and editing. For example: Line 12: Women 5-7 month's [should be months] pregnant with access to a phone Incomplete sentence: The sample was randomized and stratified by gestational age, parity, age of woman, and ownership of phone in stat using the sample command 2. Line 20 – recommend updating w latest NFHS 3. Table 2 should have a detailed footnote describing what regressions were conducted, sample sizes, etc.
--

VERSION 2 – AUTHOR RESPONSE

	Reviewer comment	Author responses
1	Thank you for revising the "Patient and Public involvement" section. However, please note that information regarding the sampling and/or recruitment should not be included in this section (please move this information to the methods section). The Patient and Public involvement" section should provide a brief response to the following questions: How was the development of the research question	Thank you. We have revised this section to take out sampling and recruitment. The paragraph now provides a brief response to those questions.

	and outcome measures informed by patients' priorities, experience, and preferences? How did you involve patients in the design of this study? Were patients involved in the recruitment to and conduct of the study? How will the results be disseminated to study participants? For randomised controlled trials, was the burden of the intervention assessed by patients themselves? Patient advisers should also be thanked in the contributorship statement/acknowledgements. If patients and or public were not involved, please state this.	
2	The manuscript needs close proofreading and editing. For example: Line 12: Women 5-7 month's [should be months] pregnant with access to a phone Incomplete sentence: The sample was randomized and stratified by gestational age, parity, age of woman, and ownership of phone in stat using the sample command	Thank you. The article was re-proofread and edited. Those lines were fixed.
3	Line 20 – recommend updating w latest NFHS	Thank you. We updated the statistics to NFHS-5 that was just released.
4	Table 2 should have a detailed footnote describing what regressions were conducted, sample sizes, etc.	Thank you. Table 2 has no regressions conducted, and in the paragraph explaining Table 2, the sample sizes were listed. Table 3, which has regressions, has the regressions described in paragraph explaining Table 3. The sample size is listed in the Study design and sampling section, but we have also added it to the paragraph explaining table 3.